# Third primary SARS-CoV-2 mRNA vaccines enhance antibody responses in most patients with haematological malignancies

Lucy B. Cook[1,2], Gillian O'Dell[2], Eleni Vourvou[2], Renuka Palanicawandar[2], Sasha Marks[2], Dragana Milojkovic[1,2], Jane F. Apperley [1,2], Sandra Loaiza[2], Simone Claudiani[2], Marco Bua[2], Catherine Hockings[2], Donald Macdonald[1,2], Aris Chaidos[1,2], Jiri Pavlu[2], Nichola Cooper[1,2], Sarah Fidler [3], Paul Randell[4] & Andrew J. Innes [1,2] ✉

SARS-CoV-2 infection, and resulting disease, COVID-19, has a high mortality amongst patients with haematological malignancies. Global vaccine rollouts have reduced hospitalisations and deaths, but vaccine efficacy in patients with haematological malignancies is known to be reduced. The UK-strategy offered a third, mRNA-based, vaccine as an extension to the primary course in these patients. The MARCH database is a retrospective observational study of serological responses in patients with blood disorders. Here we present data on 381 patients with haematological malignancies. By comparison with healthy controls, we report suboptimal responses following two primary vaccines, with significantly enhanced responses following the third primary dose. These responses however are heterogeneous and determined by haematological malignancy sub-type and therapy. We identify a group of patients with continued suboptimal vaccine responses who may benefit from additional doses, prophylactic extended half-life neutralising monoclonal therapies (nMAB) or prompt nMAB treatment in the event of SARS-CoV-2 infection.

Severe acute respiratory syndrome coronavirus-2 (SARS-CoV-2) infection and the resulting disease, COVID-19, first emerged in 2019, before being declared pandemic by the WHO in March 2020. Those at highest risk of severe disease and death included the elderly, and those with pre-existing health conditions including cancers[1–3]. Patients with blood cancers were defined as extremely clinically vulnerable to COVID-19 and advised to shield during first and subsequent waves of infection. In the first wave, patients with haematological malignancies (HM) had poor outcomes with mortality rates of 20–40%[4–7].

UK data during the initial Omicron wave (December 2021 to April 2022) has shown that despite high levels of community infections, hospital admissions and deaths are proportionally lower than previous waves. This change is likely multifactorial; largely driven by viral variant characteristics, population-level immune protection conferred by vaccination, and availability of antiviral and monoclonal antibody therapies, particularly in the non-hospitalised setting.

Prophylactic vaccines focus on immunisation with the spike (S) protein, the main target for neutralising antibodies. Neutralising antibodies block viral entry into host cells by preventing interaction between the spike protein receptor binding motif and the host cell angiotensin-converting enzyme-2, and vaccines were expected to be protective for alpha strains[8,9]. Commercial quantitative anti-S antibody assays are widely available, although there remains uncertainty as to what threshold of anti-S IgG titre correlates with effective viral

[1]Centre for Haematology, Faculty of Medicine, Department of Immunology and Inflammation, Imperial College London, London, UK. [2]Department of Haematology, Hammersmith Hospital, Imperial College Healthcare NHS Trust, London, UK. [3]Section of Virology, Faculty of Medicine, Department of Infectious Diseases, Imperial College London, London, UK. [4]Department of Infection and Immunity, North West London Pathology, London, UK. ✉e-mail: a.innes@imperial.ac.uk

neutralisation. Results from the RECOVERY trial confirm a higher mortality in patients hospitalised with COVID-19 where no antibody response was detectable upon admission, and show that treatment with casirivimab and imedevimab, a neutralising monoclonal antibody (nMAB) cocktail, reduced the relative risk of death by 20% during pre-omicron waves of infection[10]. In immunocompromised groups, NHS-England extended the use of casirivimab and imedevimab to those with very low antibody responses, defined as anti-S titres in the bottom 10% of the assay's detection range, on the premise that these were likely inadequate responses[11]. Subsequent emergence of omicron strains, which are resistant to casirivimab and imedevimab, limited it use to those infected with the delta variant[12]. However other, omicron-active, nMAB have shown efficacy in reducing rates of hospitalisation and death when delivered in the community early in the disease course[13], and there are emerging data supporting the prophylactic use of extended-half-life nMAB in those with inadequate vaccine responses[14].

The degree to which patients with blood cancers are afforded protection by vaccination is less clear, and likely to be heterogenous. Some patients with chronic haematological malignancies in remission on long-term treatment appear to have near-normal responses[15]. However, the OCTAVE study has reported the first 600 patients, including some with haematological malignancies, or stem cell transplant ($n = 80$), and has reported no measurable anti-Spike antibody responses at 4 weeks following second primary vaccine in 11.1% and 11.9% respectively[16]. While data are beginning to emerge on responses to subsequent vaccines[17,18], granularity on the heterogeneity of this patient group remains unknown.

Patients with blood cancers were amongst the highest priority group to receive SARS-CoV-2 vaccinations in the UK, with most patients receiving either the mRNA-based Pfizer-BioNTech-BNT162b2 or adenoviral-based AstraZeneca-ChAdOx1-S/nCoV-19. Both require two vaccines to complete the primary course, and in the UK, as an urgent public health measure, the second primary vaccine was delayed to a 12-week interval for many. Based on the early evidence of sub-optimal antibody responses to two vaccines in some patients, the MHRA approved the use of a third primary mRNA-based vaccine as a reinforcing dose, given at least eight weeks after completion of the second primary dose, and distinct from the booster dose, which was additionally available 8 weeks after completion of the primary course.

Reflecting the emerging real-world data of suboptimal vaccine responses and continued vulnerability of high-risk groups, a national outpatient treatment programme was established in the UK to deliver antiviral or nMAB to at risk individuals, including those with haematological malignancies, in central hubs. Currently, the programme prioritises oral antivirals over nMABs, and makes no distinction of treatment preference based on vaccine status or serological responses.

Here, we report real-world vaccine-induced SARS-CoV-2 antibody responses in patients with haematological malignancies from the MARCH (Monitoring Adaptive Responses to COVID-19 vaccines in Haematology) study. This is a retrospective observational study with data drawn from the MARCH research database. The aim of the MARCH study is to determine adaptive immune responses to COVID-19 vaccination in patients with blood disorders using real-world data. The research database includes patients 18 years or older with a history of a blood disorder (including lymphoma) receiving clinical care within our institution, who have undergone at least one standard of care SARS-CoV2 antibody test. In this study we report serological responses to SARS-CoV2 vaccination in SARS-CoV2-naive individuals with a haematological malignancy, excluding chronic myeloid leukaemia who have been reported by our institution to have near-normal serological responses[15].

## Results

### Patient cohort and sample history

Three hundred and eighty-one patients were included in this study, with the demographics shown in Table 1. Their first two vaccinations were AstraZeneca-ChAdOx1-S/nCoV-19 in 149 patients (39.1%) and Pfizer-BioNTech-BNT162b2 in 228 (59.8%), with 1 receiving moderna-mRNA1273, and 3 unknown. Almost all patients ($n = 160$, 98.3%) received Pfizer-BioNTech-BNT162b2 for their third primary dose. The median time between first and second vaccines was 75 [8–181] days,

**Table 1 | Patient demographics**

| | Post 1 (n = 171) | Post 2 (n = 327) | Post 3 (n = 162) | |
|---|---|---|---|---|
| **Age** | 62 [22–86] | 65 [22–89] | 66 [22–89] | P = 0.004 |
| **Sex** | | | | P = 0.858 |
| Male | 106 (62.0%) | 201 (61.5%) | 96 (59.3%) | |
| Female | 65 (38.0%) | 126 (38.5%) | 66 (40.7%) | |
| **Disease** | | | | P = 0.644 |
| Acute leukaemia/MDS | 2 (1.2%) | 10 (3.1%) | 5 (3.1%) | |
| Allograft <1 y | 11 (6.4%) | 11 (3.4%) | 5 (3.1%) | |
| Allograft >1 y | 34 (19.9%) | 64 (19.6%) | 29 (17.9%) | |
| CLL | 18 (10.5%) | 41 (12.5%) | 19 (11.7%) | |
| Lymphoma (B cell) | 22 (12.9%) | 45 (13.8%) | 23 (14.2%) | |
| Lymphoma (Hodgkin) | 9 (5.3%) | 9 (2.8%) | 4 (2.5%) | |
| Lymphoma (PTLD) | 4 (2.3%) | 3 (0.9%) | 3 (1.9%) | |
| MPN (ET) | 24 (14%) | 50 (15.3%) | 32 (19.8%) | |
| MPN (Myelofibrosis) | 16 (9.4%) | 28 (8.6%) | 14 (8.6%) | |
| MPN (PV) | 11 (6.4%) | 36 (11%) | 19 (11.7%) | |
| Plasma Cell Dyscrasia | 15 (8.8%) | 20 (6.1%) | 7 (4.3%) | |
| Other (MPN other/ Lymphoma other) | 5 (2.9%) | 10 (3.1%) | 2 (1.2%) | |
| **Treatment** | | | | P = 0.781 |
| BTKi | 10 (5.8%) | 13 (4.0%) | 8 (4.9%) | |
| Chemo (<6 m) | 21 (12.3%) | 30 (9.2%) | 11 (6.8%) | |
| Hydroxycarbamide | 19 (11.1%) | 53 (16.2%) | 33 (20.4%) | |
| Interferon | 7 (4.1%) | 9 (2.8%) | 5 (3.1%) | |
| JAKi | 11 (6.4%) | 21 (6.4%) | 11 (6.8%) | |
| None | 24 (14.0%) | 59 (18.0%) | 27 (16.7%) | |
| PTx-Chemo | 3 (1.8%) | 3 (0.9%) | 1 (0.6%) | |
| PTx-Immunosuppression | 19 (11.1%) | 27 (8.3%) | 17 (10.5%) | |
| PTx-Off Immunosuppression | 17 (9.9%) | 32 (9.8%) | 10 (6.2%) | |
| PTx-TKI | 5 (2.9%) | 11 (3.4%) | 6 (3.7%) | |
| R/O-Chemo (<6 m) | 18 (10.5%) | 37 (11.3%) | 20 (12.3%) | |
| R/O-Chemo (6–24 m) | 5 (2.9%) | 5 (1.5%) | 2 (1.2%) | |
| R/O-Chemo (>24 m) | 4 (2.3%) | 11 (3.4%) | 5 (3.1%) | |
| Venetoclax | 1 (0.6%) | 9 (2.8%) | 4 (2.5%) | |
| Other | 7 (4.1%) | 7 (2.1%) | 2 (1.2%) | |
| **Response** | | | | |
| Detectable anti-S | 81 (47.4%) | 239 (73.1%) | 140 (86.4%) | P < 0.001 |
| Anti-S titre >568 BAU/ml | 3 (1.8%) | 63 (19.3%) | 95 (58.6%) | P < 0.001 |

*MDS* myelodysplastic syndrome, *CLL* chronic lymphocytic leukaemia, *PTLD* post-transplant lymphoproliferative disorder, *MPN* myeloproliferative neoplasm, *ET* essential thrombocythemia, *BTKi* burton tyrosine kinase inhibitor, *Chemo* cytotoxic chemotherapy, *JAKi* Jak-stat inhibitor, *PTx* post allogeneic stem cell transplant, *TKI* tyrosine kinase inhibitor, *R/O* rituximab or obinotuzu-mab, *BAU* binding antibody units.

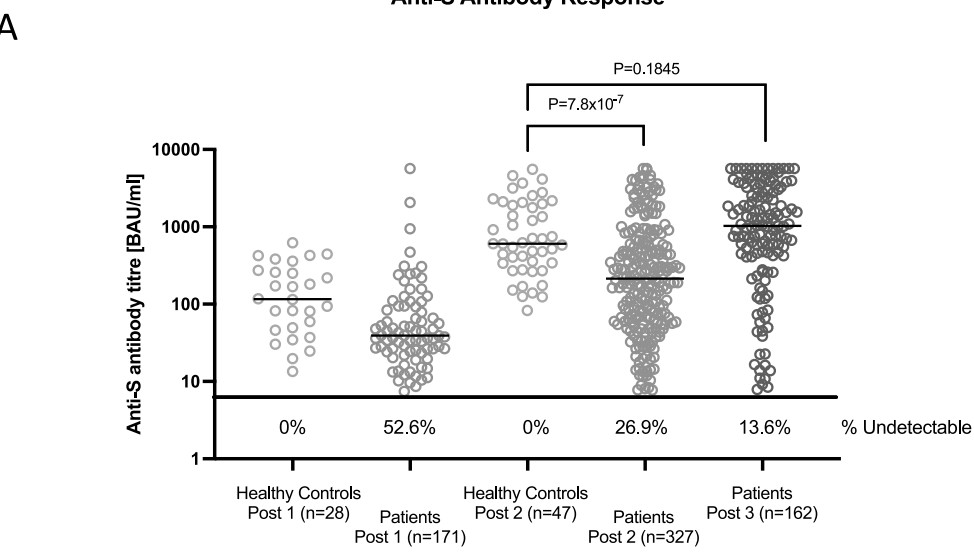

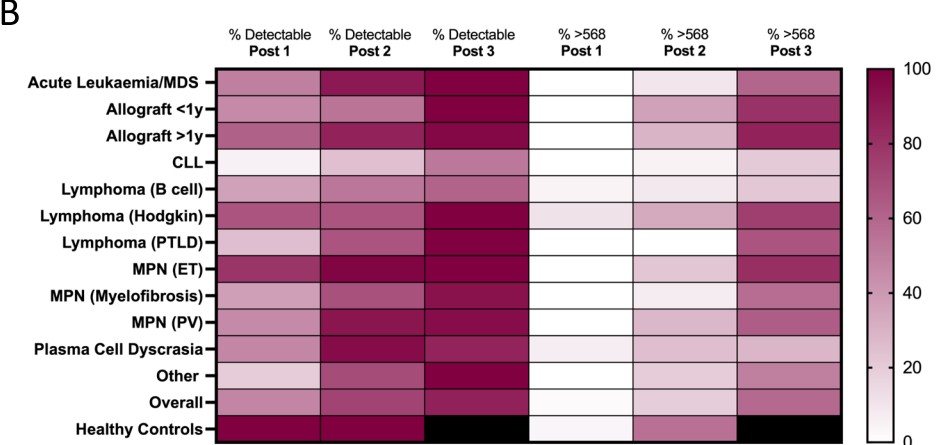

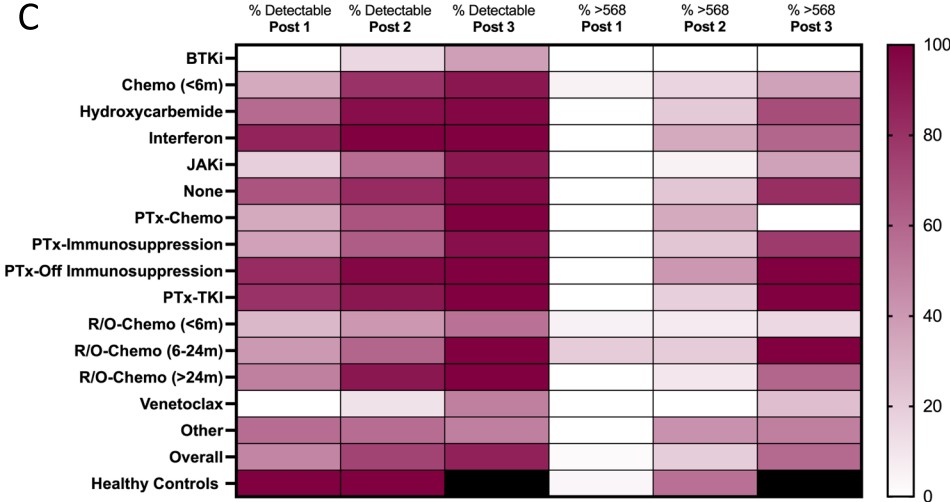

**Fig. 1 | Serological responses to SARS-CoV-2 vaccination in patients with haematological malignancies. A** Anti-S antibody titres following 1, 2 and 3 vaccines dose in patients, and following 1 and 2 in healthy controls, bar denotes median. **B** Heatmap showing the percentage of patients with detectable anti-S antibodies (>7.1 BAU/ml) and antibody response greater than the bottom 10% of the antibody range (>568 BAU/ml) stratified by disease type. **C** Heatmap stratified by treatment. BAU binding antibody units, MDS myelodysplastic syndrome, CLL chronic lymphocytic leukaemia, PTLD post-transplant lymphoproliferative disorder, MPN myeloproliferative neoplasm, ET essential thrombocythemia, PV polycthaemia vera, BTKi burton tyrosine kinase inhibitor, Chemo cytotoxic chemotherapy, JAKi Jak-stat inhibitor, PTx post allogeneic stem cell transplant, TKI tyrosine kinase inhibitor, R/O rituximab or obinotuzumab. Paired analysis by Mann–Whitney U test (two sided).

and between second and third was 187 [114–300] days. Antibody results were available after one, two and three vaccines for 171, 327 and 162 patients respectively, with a median time between vaccination and antibody test of 57 [11–131] days, 49 [5–187] days and 50 [9–117] days post first, second and third doses respectively (Supplementary Fig. 1). At each of the three timepoints evaluated, there were no significant differences in the representation of patient diseases or treatment groups (Table 1), although there was a small but significant difference in median age (62, 65, 66 years post first, second and third respectively). Healthy controls have been previously reported by our group and were sampled at day 21 (n = 13) and 49 (n = 15) post first dose and day 21 (n = 24) and 90 (n = 23) post second dose[15].

### Impact of HM and treatment on serological responses

While all healthy controls had detectable anti-S antibodies after the first vaccination, this was not true in patients with haematological malignancies. There was however an incremental increase in the proportion of patients with detectable responses after the first, second and third doses (47.4% 73.1% and 86.4% respectively P < 0.001, Table 1, Fig. 1A). Responses were not uniform however, and whilst overall, detectable responses were seen in 86.4% of patients after 3 doses, only 52.6% of those with CLL and 60.9% of those with B-cell lymphoma had measurable anti-S antibodies (Fig. 1B, Supplementary Fig. 2). There were also significant differences when patients were stratified by treatment, with detectable responses after 3 doses in only 37.5% of those treated with Bruton's tyrosine kinase (BTK) inhibitors, 55.0% of those receiving anti-CD20 monoclonal antibody-containing regimens (rituximab or obinutuzumab) in the past 6 months, and 50% of those on the bcl-2 inhibitor venetoclax (Fig. 1C, Supplementary Fig. 3). There was a clear divergence in response to the third dose, with some groups of patients with previously reported poor responses to 2 doses, such as those on JAK-STAT inhibitors[19] being partially rescued by the 3rd dose, (detectable anti-S antibodies after 1, 2 and 3 doses of 18.2%, 57.1% and 90.9% respectively) whilst other groups, such as those on BTK inhibitors, showed only modest increase in antibody detection with increasing number of doses (anti-S antibodies after 0%, 15.4% and 37.5% after 1, 2 and 3 doses respectively) (Fig. 1C, Supplementary Fig. 3).

### Impact of HM and treatment on antibody titre

For patients with detectable anti-S antibodies after the second vaccine, the median titre was significantly lower than healthy controls (patients 214.0 [7.8–5680] BAU/ml, controls 603.6 [82.8–5531] BAU/ml, P < 0.001, Fig. 1A). However anti-S titres following a third primary vaccine in patients was not significantly different from the post second anti-S titre observed in the controls (median for patients after three doses 1026 [7.89-5680] BAU/ml and controls after two doses 603.6 [82.81–5531] BAU/ml, P = 0.1845, Fig. 1A).

While only 19.3% of patients had anti-S tires above 568 BAU/ml (bottom 10% of antibody detection range) compared to 55.3% of healthy controls following two vaccinations, 58.6% of patients achieve this after the third primary dose. Again, these responses were not uniform, with only 21.1% of those with CLL, 21.7% of those with B-cell lymphoma, and 28.6% of those with plasma cell dyscrasias crossing this threshold (Fig. 1B, Supplementary Fig. 2). Importantly no patients treated with a BTK inhibitor mounted responses above 568 BAU/ml, while only 15.0% of those receiving rituximab- or Obinutuzumab-containing regimens in the past 6 months, 25.0% of those receiving venetoclax, 36.4% receiving other cytotoxic chemotherapy and 36.4% receiving JAK-STAT inhibitors achieved titres over 568 BAU/ml (Fig. 1C, Supplementary Fig. 3).

### Impact of vaccine type on serological response

While there were significant disparities between serological responses after 2 vaccines between those receiving ChAdOx1-S/nCoV-19 or BNT162b2, these were a overcome by responses to the third primary mRNA-based vaccine, after which no significant difference were seen in either percentage of patients mounting a detectable response (91.9% vs 82.8%, P = 0.102), or the median titre of those responding (1299 BAU/ml vs 930 BAU/ml, P = 0.560, Supplementary Fig. 4).

## Discussion

The MARCH study was designed to report real-world serological responses to SARS-CoV-2 vaccination in a large number of patients with haematological malignancies. There are of course biases in the nature of this study design, including the heterogeneity of underlying disease and treatments, and potentially, selection bias for those patients who continued to attend for appointments during pandemic where phlebotomy was performed. Nevertheless, it captures the most at-risk patients groups and provides insights into the quantitative nature of vaccine responses in a specific vulnerable patient group. Whilst the healthy controls and the patients with malignancies were not perfectly matched, they provide context to the stark difference in antibody responses.

Following the first two vaccines, a substantial proportion of patients with haematological malignancies fail to mount any detectable serological response, and many of those who do, have only low-level titres with unknown degree of protection. Importantly, a large proportion of patients were effectively salvaged following the third primary vaccine, with antibody levels similar to those of the healthy controls who had received two doses. While some of the enhanced responses to the third dose may be explained by a longer period since treatment completion, for others, such as those on continuous therapy, there is no obvious clinical explanation. There does however remain a significant subset of vulnerable patients, primarily those with CLL, B-cell lymphomas, myelofibrosis, and those on specific therapies (B-cell depleting monoclonal antibodies, BTK-, BCL2- and JAK-STAT-inhibitors) who continue to have suboptimal vaccine responses even after three doses and likely remain vulnerable to severe infection, and who may benefit most from pre-emptive or prophylactic nMAB therapies.

Current NHS-England and CDC guidance tends to group patients with haematological malignancies together as a single cohort, and these data suggests this is not a homogenous group, and whilst oral antivirals may be reasonable treatments in some groups, it suggests that others with no, or low-level, sero-protection following vaccination may benefit more from neutralising monoclonal antibodies in first-line therapy in the event of SARS-CoV-2 infection, a provision that has already been made for those with solid organ transplantation in the UK. The emerging trends also suggest that sup-optimal responses can be enhanced with additional primary vaccine doses, supporting the ongoing strategy for 4th and subsequent doses in this patient group, but others may fail to prime immune responses despite serial vaccination, and may be better managed with extended-half-life nMAB.

## Methods

### Ethical approval

The MARCH research database was approved by the Health Research Authority, Research Ethics Committee (reference: 21/NI/0176, IRAS project ID 299969, sponsored by Imperial College Healthcare NHS Trust, approved by Office for Research Ethics Committees Northern Ireland), and complies with all relevant ethical regulations. Data used for research purposes was de-identified before export to the research environment (pseudoanonymised minimal dataset comprising database unique identifier, age, sex, disease, treatment, vaccine type, serological response, time from vaccine to test). Under the terms of approval, for a COVID-19 purpose, informed consent was not required under direction of the UK Secretary of State. The study was undertaken in accordance with the criteria set by the Declaration of Helsinki. There was no compensation for participation.

## Patient characteristics and healthy volunteers

SARS-CoV-2 antibodies are reported on 381 patients with blood cancers who have received prophylactic SARS-CoV-2 vaccinations. SARS-CoV-2 antibodies measured during routine care of patients attending the Department of Haematology, Imperial College Healthcare NHS Trust, London since March 2021 were collated in the MARCH (Monitoring Adaptive Responses to COVID-19 vaccines in Haematology) database and correlated with demographic and vaccination data from electronic health records. Serological responses were measured at routine clinical visits, and not at mandated timepoints pre-determined by a study protocol (Supplementary Fig. 1). Data cut off for inclusion in this study was 31 January 2022. To determine the vaccine response in SARS-CoV-2-naive individuals, those with a history of prior known SARS-CoV-2 infection (PCR-proven), or the presence of an anti-nucleocapsid protein antibody at the time of testing were excluded from this analysis. Twenty-eight healthy controls aged between 25–66 years with no prior history of SARS-CoV-2 infection, who received Pfizer-BioNTech-BNT162b2 ($n = 25$) or AstraZeneca-ChAdOx1-S/nCoV-19 ($n = 3$) were recruited in the prospective COVAX study and have previously been reported[15]. Healthy controls underwent testing at 21 ($n = 13$) and 49 ($n = 15$) days post first dose and 21 ($n = 24$) and 90 ($n = 23$) days post vaccine 2, and post first and second vaccine results were amalgamated.

## SARS-CoV-2 IgG assay

Sera from patients were tested for the presence of nucleocapsid protein (NP) antibodies and quantitative spike (S) IgG antibodies using the chemiluminescent microparticle immunoassay Abbott SARS-CoV-2 IgG II quant assay on the Abbott Architect i System according to manufacturers' instructions (Abbott 6S60). The assay is designed to detect IgG antibodies, including neutralising antibodies, to the receptor binding domain (RBD) of the S1 subunit of the spike protein. Results are reported in Binding Antibody Units/mL (BAU/mL) calculated by multiplying the Abbott AU/mL by a conversion factor of 0.142, as described in the manufacturer's instructions, on a scale of undetectable (<7.1) to >5680 BAU/mL. Anti-S in the lowest 10% of the antibody detection range were anti-S IgG < 568 BAU/mL.

## Statistical analysis

Microsoft Access Database 2016 was used for data collection. Microsoft Excel 2016, SPSS Statistics version 25, and GraphPad Prism versions 9.3.1 and 9.4.1 were used for data analysis. Categorical variables are compared using Chi squared test (two sided). Continuous variables are compared using Mann-Whitney U test (two sided) when comparing two variables with non-standard distribution (anti-S antibody titres) or one-way ANOVA for more than 2 group of variables are compared (e.g. Patient age).

## Reporting summary

Further information on research design is available in the Nature Portfolio Reporting Summary linked to this article.

## Data availability

The source data for production of figures are provided as a Source Data file. In accordance with ICJME guidelines, Individual participant data that underlies the results reported in this article, after de-identification, will be shared, beginning 9 months and ending 36 months following article publication. Data will be shared with investigators whose proposed use of the data has been approved by an independent review committee identified for this purpose, for meta-analysis purposes. Proposals may be submitted to the corresponding author within 36 months of article publication. Manufacturers assay protocol can be provided upon request. The study protocol for the research database is available upon request from the corresponding author. Source data are provided with this paper.

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

## Acknowledgements

This study was supported by the Imperial College NIHR BRC. L.C. is in part supported by the Imperial College NIHR BRC. We are grateful for the statistical support provided by Dr. Richard Szydlo at Imperial College

London, and to the clinical, nursing, allied healthcare, and administrative staff of the department of haematology, Hammersmith Hospital, London, and administrative support of the Medical Directors Office Team, at Imperial College Healthcare NHS Trust.

## Author contributions

L.C., G.O. and A.I. designed and oversaw the study. E.V. supported ethics approval and data management. R.P., S.M., D.M., J.A., S.L., S.C., M.B., C.H., D.M., A.C., J.P. and N.C. provided data. L.C., G.O., R.P., S.M., D.M., J.A., S.C., M.B., C.H., D.M., A.C., J.P., N.C. and A.I. were involved in the clinical care of patients included in the study. D.M., S.C., J.A. and A.I. provided normal control data. P.R. oversaw the laboratory work. S.F. provided research direction and discussion. L.C. and A.I. analysed that data and wrote the manuscript. All authors approved the final manuscript version.

## Competing interests

The authors declare no competing interests.
