## [Peer Review File · Nature Communications]

Third primary SARS-CoV-2 mRNA vaccines enhance antibody responses in most patients with haematological malignanciesThis manuscript has been previously reviewed at another journal that is not operating a transparent peer review scheme. This document only contains reviewer comments and rebuttal letters for versions considered at *Nature Communications*.

REVIEWER COMMENTS

Reviewer #2 (Remarks to the Author):

The authors quantify serological responses following covid vaccines in a cohort of 381 patients with haematological malignancies attending routine haematology outpatient clinics. They report suboptimal responses following two primary vaccines, with significantly enhanced responses following the third primary dose. These responses however are less in patients with Lymphoid neoplasms and those on anti B cell and BTK drugs.

The strength of the real world data presented in this study.

The novelty is modest though, as multiple groups have shown similar findings. Lower responses in these lymphoid subgroups have been seen in many different studies.

Reviewer #3 (Remarks to the Author):

Cook and colleagues profiled spike-specific antibodies in 381 hematologic cancer patients after SARS-CoV-2 vaccination, including 162 patients after a 3rd dose of vaccine. The authors show that while 72.8% of patients have a detectable spike-specific antibody response after the 2nd dose, it increases to 87% after the 3rd dose. And while hematologic cancer patients had anti-S titers that was decreased compared to healthy controls after the 2nd dose, these titers become comparable to healthy controls after a 3rd vaccination. The strength of this paper is that it is that it is a large study of hematologic cancer patients, and has the numbers to study different types of hematologic cancer patients with different treatment types. While certainly relevant to the current COVID-19 pandemic, the study is not particularly novel, and lacks neutralizing data. For example, Greenberger et al, Cancer Cell 2021 demonstrated in 49 patients with B cell malignancies, that a 3rd dose of vaccine boosted anti-Spike antibodies. Fendler et al, Lancet 2022, reported that a 3rd dose of vaccine boosted neutralizing antibody responses in 79 hematologic cancer patients, including those targeting both omicron and delta. Ehmsen et al, Cancer Cell 2022 looked at 316 hematologic cancer patients after a 3rd vaccination. Thus, as is, there believe the paper lacks the novelty for Nature Cancer submission.

Reviewer #4 (Remarks to the Author):

SUMMARY

In this article, Cook et al sought to characterize serological responses to COVID-19 vaccination after primary series and following a boost in patients with hematologic malignancies. They measured SARS-CoV-2-Spike binding antibody responses from available samples in a group of 381 patients with a variety of hematologic malignancies, following vaccination with a primary series consisting of the AstraZeneca ChAdOx1 vaccine or the Pfizer-BioNTech BNT162b2 vaccine, and boosting with an additional dose of the Pfizer vaccine.

The authors report that all healthy controls had detectable anti-S antibodies after the first vaccination, that approximately 52.6% of those with hematologic malignancies responded after the first vaccination, and that in this group, response rates rose to 72.8% and 87.0% after the second and third doses, respectively. The median antibody response in this group after the third dose appeared to be on par with or slightly higher than antibody responses in healthy controls after 2 doses, although a significant number of individuals in the malignancy group continued to have poor responses. These data suggest that a booster dose can overcome the immune deficit seen after the primary vaccine series in these patients, although some individuals and/or particular diagnoses may be more refractory to boosting.

Overall, the manuscript is well organized and reasonably clearly written, although a few clarifications are requested (see major comments). The authors address an issue of major importance for individuals diagnosed with hematologic malignancies, and their data suggest that in most individuals experiencing hematologic malignancies, a third dose of the Pfizer vaccine can

result in measurable antibodies responses.

Although there are now numerous articles addressing immunologic responses to COVID-19 vaccines in patients with hematologic malignancies, relatively few address the impact of boosters. One further strength of this effort is the relatively large number of individuals enrolled, and the fairly broad spectrum of illnesses include. Lastly, the authors include information pertaining to 2 different primary vaccine regimens, has the potential to provide important insight. However, the impact of this manuscript is affected by the very inconsistent sampling, small numbers of participants for some disease states, and by some aspects of the presentation and analysis which are detailed below. The authors likely cannot do anything to change cohort and sampling details but should attempt to address the comments below in a revised manuscript.

MAJOR COMMENTS

1. In figure 1A, the data in the last group of patients (patients post 3 (N equals 162)) would appear to have a maximum value cutoff, possibly representing the limits of the linear range of the assay (this is also seen in supplemental figure 1). This might affect the statistical comparison between this group and the other groups included. For samples exceeding the upper limit of the assay, dilutions should be performed and the samples rerun and corrected values provided.
2. In figure 1B, results by diagnosis and treatment are presented as a heat map. However, for some disease conditions there are very few data available. For example, for lymphoma there appear to be only four post vaccine 1 samples, three post vaccine 2 samples and 3 post vaccine 3 samples. The heat map for this condition shows progression from light blue to dark blue, but it is hard to gauge the actual response level/percentage in this group, or the significance of the results. To convey this information more concretely, the authors should consider a numeric representation of response level along with confidence intervals.
3. Two different primary regimens were used in the original vaccine series in this cohort. The authors should provide a breakdown of immune responses by primary vaccine regimen
4. In reviewing figure 1A, the post V3 group has a distinct group of around 20 patients who failed to mount antibody responses on par with even the lowest of the normal controls. it would be helpful, perhaps in the supplementary information to detail the underlying conditions and treatments in this group.

MINOR COMMENTS

1. In the results section (lines 94-104) the authors provide some details on the number of patients enrolled, vaccinated, and tested for antibodies, as well as information time interval between vaccination and antibody test. In the subsequent paragraph the authors refer to results in "healthy controls" but no information is provided on how many healthy controls were included, or any information on vaccination, testing and timing. Sufficient detail is presented in the supplementary material, but would suggest at least providing the number of healthy controls and an indication that they are separate from the 381 patients with malignancies in the main manuscript.
2. Line 63 typographical error: Imdevimab
3. Line 208 Figure legend typographical error: "greater above

RESPONSE TO REVIEWER COMMENTS

Reviewer #2 (Remarks to the Author):

The authors quantify serological responses following covid vaccines in a cohort of 381 patients with haematological malignancies attending routine haematology outpatient clinics. They report suboptimal responses following two primary vaccines, with significantly enhanced responses following the third primary dose. These responses however are less in patients with Lymphoid neoplasms and those on anti B cell and BTK drugs.

The strength of the real world data presented in this study.

The novelty is modest though, as multiple groups have shown similar findings. Lower responses in these lymphoid subgroups have been seen in many different studies.

We thank the reviewer for these comments, but feel this work does remain novel because of its size and data granularity. While there are independent small cohorts reporting in specific diseases, our data provides a context with both normal controls, and the ability to directly compare between different subsets of haematological malignancy and treatment groups with a single test.

Reviewer #3 (Remarks to the Author):

Cook and colleagues profiled spike-specific antibodies in 381 hematologic cancer patients after SARS-CoV-2 vaccination, including 162 patients after a 3rd dose of vaccine. The authors show that while 72.8% of patients have a detectable spike-specific antibody response after the 2nd dose, it increases to 87% after the 3rd dose. And while hematologic cancer patients had anti-S titers that was decreased compared to healthy controls after the 2nd dose, these titers become comparable to healthy controls after a 3rd vaccination. The strength of this paper is that it is that it is a large study of hematologic cancer patients, and has the numbers to study different types of hematologic cancer patients with different treatment types. While certainly relevant to the current COVID-19 pandemic, the study is not particularly novel, and lacks neutralizing data. For example, Greenberger et al, Cancer Cell 2021 demonstrated in 49 patients with B cell malignancies, that a 3rd dose of vaccine boosted anti-Spike antibodies. Fendler et al, Lancet 2022, reported that a 3rd dose of vaccine boosted neutralizing antibody responses in 79 hematologic cancer patients, including those targeting both omicron and delta. Ehmsen et al, Cancer Cell 2022 looked at 316 hematologic cancer patients after a 3rd vaccination. Thus, as is, there believe the paper lacks the novelty for Nature Cancer submission.

We thank the reviewer for their comments. Regarding the referenced papers, Greenberger et al Cancer Cell 2021 reports predominantly responses to second vaccines. We agree that Fender et al Lancet 2022 and Ehmsen et al Cancer Cell 2022 report some third vaccine responses and have now made reference to these in the revised manuscript. Nevertheless, we would re-iterate the points above that these data lack the granularity of our manuscript.

Reviewer #4 (Remarks to the Author):

SUMMARY

In this article, Cook et al sought to characterize serological responses to COVID-19 vaccination after primary series and following a boost in patients with hematologic malignancies. They measured SARS-CoV-2-Spike binding antibody responses from available samples in a group of 381 patients with a variety of hematologic malignancies, following vaccination with a primary series consisting of the AstraZeneca ChAdOx1 vaccine or the Pfizer-BioNTech BNT162b2 vaccine, and boosting with an additional dose of the Pfizer vaccine.

The authors report that all healthy controls had detectable anti-S antibodies after the first vaccination, that approximately 52.6% of those with hematologic malignancies responded after the first vaccination, and that in this group, response rates rose to 72.8% and 87.0% after the second and third doses, respectively. The median antibody response in this group after the third dose appeared to be on par with or slightly higher than antibody responses in healthy controls after 2 doses, although a significant number of individuals in the malignancy group continued to have poor responses. These data suggest that a booster dose can overcome the immune deficit seen after the primary vaccine series in these patients, although some individuals and/or particular diagnoses may be more refractory to boosting.

Overall, the manuscript is well organized and reasonably clearly written, although a few clarifications are requested (see major comments). The authors address an issue of major importance for individuals diagnosed with hematologic malignancies, and their data suggest that in most individuals experiencing hematologic malignancies, a third dose of the Pfizer vaccine can result in measurable antibodies responses.

Although there are now numerous articles addressing immunologic responses to COVID-19 vaccines in patients

with hematologic malignancies, relatively few address the impact of boosters. One further strength of this effort is the relatively large number of individuals enrolled, and the fairly broad spectrum of illnesses include. Lastly, the authors include information pertaining to 2 different primary vaccine regimens, has the potential to provide important insight. However, the impact of this manuscript is affected by the very inconsistent sampling, small numbers of participants for some disease states, and by some aspects of the presentation and analysis which are detailed below. The authors likely cannot do anything to change cohort and sampling details but should attempt to address the comments below in a revised manuscript.

MAJOR COMMENTS

1. In figure 1A, the data in the last group of patients (patients post 3 (N equals 162)) would appear to have a maximum value cutoff, possibly representing the limits of the linear range of the assay (this is also seen in supplemental figure 1). This might affect the statistical comparison between this group and the other groups included. For samples exceeding the upper limit of the assay, dilutions should be performed and the samples rerun and corrected values provided.

We thank the reviewer for their comments. They have correctly observed that the assay is reported within an analytical measuring interval optimised for performance for linearity, precision, and bias. This results in a maximal reportable value of 5680 BAU/mL. While this range can be increased to an extended reporting interval by dilution, there continues to be a maximal reporting value recommended as per the manufacturer's instructions, so capped values are likely to persist even with a further dilution. We therefore addressed the most statistical robust way to process these comparisons, and given that the data are non-parametric, we have now amended the analysis to pairwise Mann Whitney U tests between healthy control post 2 and patient group post 2, and healthy control post 2 and patient group post 3 (Notably similar results are observed with a Kruskal-Wallis test with multiple comparisons, which is a non-parametric ANOVA equivalent, but given that the critical comparison are between two groups, a Mann-Whitney U test is more suitable) – This change has been reflected in the methods and legend. Most importantly however, given that those results >5680 could only be 5680 or greater, increasing these values by dilution would neither influence the median value nor influence the Mann-Whitney U test, therefore have no impact in this context. Of note, we are grateful to Dr Richard Szydlo, medical statistician for his discussion relating to this, and have added an acknowledgement to in the revised manuscript.

2. In figure 1B, results by diagnosis and treatment are presented as a heat map. However, for some disease conditions there are very few data available. For example, for lymphoma there appear to be only four post vaccine 1 samples, three post vaccine 2 samples and 3 post vaccine 3 samples. The heat map for this condition shows progression from light blue to dark blue, but it is hard to gauge the actual response level/percentage in this group, or the significance of the results. To convey this information more concretely, the authors should consider a numeric representation of response level along with confidence intervals.

We agree that further explanation of the data would be helpful, but do not want to negatively impact the aesthetic of the main figure, but are of course happy to open provide this data. We have altered the colour scheme of figure 1a to try and improve clarity and have added 2 additional supplemental figures that show the raw values overlaid on each cell of the heatmap (supplemental figures 2a and 3a), as well as a table detailing the 95% confidence intervals (supplemental figures 2b and 3b).

Of note, I think the reviewer may have made reference to Lymphoma (PTLD), with regard to the number of patients post vaccine being small (n=4, n=3, n=3 post 1, 2 and 3 vaccines respectively), and we would clarify that this applies to a specific lymphoma (lymphoma (PTLD)) subset only, and not the majority of B-cell lymphoma (lymphoma (B Cell) patients which comprise 22, 45 and 23 patients post 1, 2 and 3 vaccine respectively. We discussed as group whether these patients (and equally other groups with small numbers) should be merged with other similar groups; for example adding Lymphoma (PTLD) to Lymphoma (B cell), because there are by definition all B-cell lymphomas. However, we felt that these groups are significantly different (by nature of their ongoing immunosuppression for solid organ transplant), that there was merit in segregating them, if only to highlight the limited data in these subgroups. We fully agree with the reviewer that we should make reference to the small numbers in these sets, and hope by including the confidence intervals this has been achieved.

3. Two different primary regimens were used in the original vaccine series in this cohort. The authors should provide a breakdown of immune responses by primary vaccine regimen

The reviewer raises an important point, and we have now presented serological responses after each vaccine stratified by the initial 2 vaccines type (ChAdOx1-S/nCoV-19 or Pfizer-BioNTech-BNT162b2). This is now presented in supplemental figure 4 and referred directly to this in the revised manuscript text. Critically, while there are disparities in response after second vaccines between those receiving ChAdOx1-S/nCoV-19 or Pfizer-BioNTech-BNT162b2, all patients had an mRNA based third vaccine (almost all of which (98.3%) were Pfizer-BioNTech-BNT162b2) and this negates those differences.

4. In reviewing figure 1A, the post V3 group has a distinct group of around 20 patients who failed to mount antibody responses on par with even the lowest of the normal controls. it would be helpful, perhaps in the supplementary information to detail the underlying conditions and treatments in this group.

The reviewer has correctly observed that there is a group of patients with responses below the lowest anti-S titre of the healthy control (<82 BAU.mL). We have included a heatmap below separated by disease and treatment, showing which patients groups fall into this category. Importantly these groups mirror those who fail to mount responses, and those who fail to mount responses > 568 BAU/ml. Therefore, while we agree that this is interesting, it does not change the risk groups identified described in the manuscript, and overall we feel it does not add merit to the manuscript.

MINOR COMMENTS

1. In the results section (lines 94–104) the authors provide some details on the number of patients enrolled, vaccinated, and tested for antibodies, as well as information time interval between vaccination and antibody test. In the subsequent paragraph the authors refer to results in "healthy controls" but no information is provided on how many healthy controls were included, or any information on vaccination, testing and timing. Sufficient detail is presented in the supplementary material, but would suggest at least providing the number of healthy controls and an indication that they are separate from the 381 patients with malignancies in the main manuscript.

2. Line 63 typographical error: Imdevimab – Amended

3. Line 208 Figure legend typographical error: "greater above – Amended

REVIEWERS' COMMENTS

Reviewer #4 (Remarks to the Author):

In the revised version of this manuscript, the authors have largely addressed the major comments from my previous review. They have performed a nonparametric analysis of antibody responses to address the issue of comparability between groups involving data where actual responses exceed the dynamic range of the assay. They have provided additional figures in the supplement showing raw data corresponding to the heat map figures to allow the reader a deeper insight into the actual numbers and likely statistical power of this analysis. They have also provided a figure presenting serological responses by vaccine type as requested. They have also provided a new figure showing response rates by diagnosis that provides sufficient detail on the specific diagnoses associated with lack of response to vaccination. I have no further concerns regarding the issues raised in my prior review.